# Sleep need driven oscillation of glutamate synaptic phenotype

Kaspar E Vogt[1], Ashwinikumar Kulkarni[2], Richa Pandey[3], Mantre Dehnad[3], Genevieve Konopka[2], Robert W Greene[1,2,3]*

[1]International Institute of Integrative Sleep Medicine, University of Tsukuba, Tsukuba, Japan; [2]Department of Neuroscience, Peter O'Donnell Brain Institute, University of Texas Southwestern Medical Center, Dallas, United States; [3]Department of Psychiatry, Peter O'Donnell Brain Institute, University of Texas Southwestern Medical Center, Dallas, United States

## eLife Assessment

This **important** study showing that sleep deprivation increases functional synapses while depleting silent synapses supports previous findings that excitatory signaling increases during wakefulness. This manuscript focuses in particular on AMPA/NMDA ratios. An interesting, although speculative, aspect of the manuscript is the inclusion of a model for the accumulation of sleep needs that is based upon the MEF2C transcription factor but also links to the sleep-regulating SIK3-HDAC4/5 pathway. The authors have clarified some questions raised in the previous review, rendering this a **solid** piece of work that poses questions for future studies.

*For correspondence:
robertw.greene@
utsouthwestern.edu

## Abstract

Sleep loss increases AMPA-synaptic strength and number in the neocortex. However, this is only part of the synaptic sleep loss response. We report an increased AMPA/NMDA EPSC ratio in frontal-cortical pyramidal neurons of layers 2–3. Silent synapses are absent, decreasing the plastic potential to convert silent NMDA to active AMPA synapses. These sleep loss changes are recovered by sleep. Sleep genes are enriched for synaptic shaping cellular components controlling glutamate synapse phenotype, overlap with autism risk genes, and are primarily observed in excitatory pyramidal neurons projecting intra-telencephalically. These genes are enriched with genes controlled by the transcription factor, MEF2c, and its repressor, HDAC4. Sleep genes can thus provide a framework within which motor learning and training occur mediated by the sleep-dependent oscillation of glutamate-synaptic phenotypes.

## Introduction

CNS control of arousal maintains waking neuronal activities needed for foraging, danger avoidance, and reproductive behaviors. Reduction of arousal to unconsciousness can gate sleep, however, the function served by this sleep, in contrast to wake function, remains enigmatic. A reasonable teleologically oriented goal is more effective future foraging, danger avoidance, and reproductive behaviors, based on procedural and episodic learning from past waking experiences. Effective learning involves memory of waking experience and underlying long-term plasticity (LTP) (**Moser et al., 1998**; **Morris et al., 2003**; **Stickgold, 2005**). At the cellular level, extensive activity-dependent LTP of cortical glutamate synapses can lead to a well-documented increase in synaptic strength of cortical glutamatergic synapses during waking. It is reflected by the increased frequency and amplitude of glutamatergic miniature synaptic currents (mEPSC) that are restored by subsequent sleep (**Liu et al., 2010**; **Torrado**

*Pacheco et al., 2021*), consistent with the synaptic homeostasis hypothesis of sleep (SHY) (*Tononi and Cirelli, 2014*).

Our recent findings indicate that during waking, not only are individual synapses strengthened (overall glutamate mEPSC amplitude in layer 2–3 pyramidal cells is increased by SD) but, also, the number of electro-physiologically active synapses is increased (increased frequency in correlation with unaltered or decreased probability of release) (*Bjorness et al., 2020*). The molecular underpinnings of these changes in glutamate synaptic strength and number were examined to determine the changes in the glutamate synaptic phenotype and what the implication(s) of these changes, if any, might be with regard to a propensity for synaptic plasticity (meta-plasticity).

We investigated sleep/wake mediated electrophysiological changes in glutamatergic synapses in the motor cortex, layer 2–3, pyramidal neurons since our previous work demonstrates wake/sleep up-regulation and down-regulation, respectively, in these cells' glutamate synapses (*Bjorness et al., 2020*). We observed a wake/sleep oscillation of AMPA/NMDA synaptic response ratio in pyramidal neurons of the motor neocortex coupled with an oscillating fraction of silent synapses (NMDAR-mediated synaptic response but no AMPAR-mediated response) in response to sleep need. We then, extended these physiological findings to include the characterization of sleep-need-related differential expression of glutamatergic synaptic shaping genes that can control the observed oscillating glutamate synapse phenotype.

## Results

### Sleep need driven, glutamate, synaptic phenotype

Three cohorts of C57 BL/6 male mice were allowed either to (1) sleep ad lib (control sleep, CS); (2) prevented from sleeping from ZT = 0–6 hr (sleep deprived, SD); or, (3) after 4 hr SD allowed recovery sleep, ad lib for 2 hr (RS). Ex-Vivo acute brain slices from the motor cortex were then prepared from these animals at ZT = 6 hr and whole cell recordings were obtained from layer 2/3 pyramidal neurons (see methods for details). Locally evoked glutamatergic EPSCs recorded at $V_{hold}$ = –90 mV had little NMDAR contribution due to $Mg^{++}$ block. These were compared in the same cell, to EPSCs evoked at $V_{hold}$ = +50 mV with little AMPAR contribution due to reduced AMPAR driving force, but, having large NMDAR currents due to voltage sensitive relief from $Mg^{++}$ blockade (NMDAR currents had reversed polarity since $E_{NMDAR}$ = ~0 mV). For each recorded neuron (one neuron/slice/animal) an average response to repeated stimuli (>10 stimuli, 100 pA, applied every 10 s) to elicit an EPSC was determined at each holding potential. A two-way ANOVA analysis showed a significant interaction between AMPA matched to NMDA EPSC response for each neuron, and sleep condition ($F_{(2, 21)}$=7.268, p<0.004; *Figure 1A, C and E*; *Figure 1—source data 1*, tabs 1&2). When considered independently, neither the effect of sleep condition nor of EPSC subtype reached significance at p<0.05 (*Figure 1C*).

To control for neuron-to-neuron EPSC variability (due to variability of stimulation-evoked afferent activation), we calculated the AMPA/NMDA ratio for three sleep conditions, CS, SD, and RS. We observed a significant difference between sleep condition cohorts (non-parametric Kruskal-Wallis test, p<0.0022). Significance tests for multiple comparisons between sleep conditions, gave CS-SD, p<0.031; SD-RS, p<0.005; and CS-RS, p<0.191 (Two-stage linear step-up procedure of Benjamini, Krieger, and Yekutieli; *Figure 1—source data 1*, tabs1&3; *Figure 1E*).

Many glutamatergic synapses on cortical pyramidal cells do not respond to activation at resting membrane potential, yet they do have an NMDAR synaptic component that can be observed at $V_{hold}$ = +50 mV. After activation by LTP-evoking stimulation, these 'silent' synapses can convert to active, non-silent synapses by virtue of LTP-induced AMPAR insertion into the post-synaptic active zone (*Liao et al., 1995*). The ratio of AMPAR component failure rate ($V_{hold}$ = –90 mv; $FR_{-90}$ for active synapses, As) matched with the same neuron's $FR_{+50}$ ($V_{hold}$ = +50 mV; for As +silent synapses, Ss) has been used to estimate the change in the fraction of Ss/As in response to LTP (*Liao et al., 1995*). We employed a similar analysis to examine the effects of SD and sleep on functional synaptic AMPAR and NMDAR composition. As with the EPSC amplitude analysis, there was a significant interaction for failure rates at $V_{hold}$ = –90 mv and $V_{hold}$ = +50 mV, and the three sleep conditions (p<0.031; $F_{(1.742, 8.708)}$=5.550; *Figure 1—source data 1*, Tabs 4&5; *Figure 1D*). Multiple comparisons (Šídák's multiple comparisons test; *Figure 1—source data 1*, Tab 5) of mean FR at $V_{hold}$ = –90 mv (for As) and $V_{hold}$ = +50 mV (for As +Ss) across sleep conditions, revealed no significant difference between $FR_{-90}$ and $FR_{+50}$ in SD

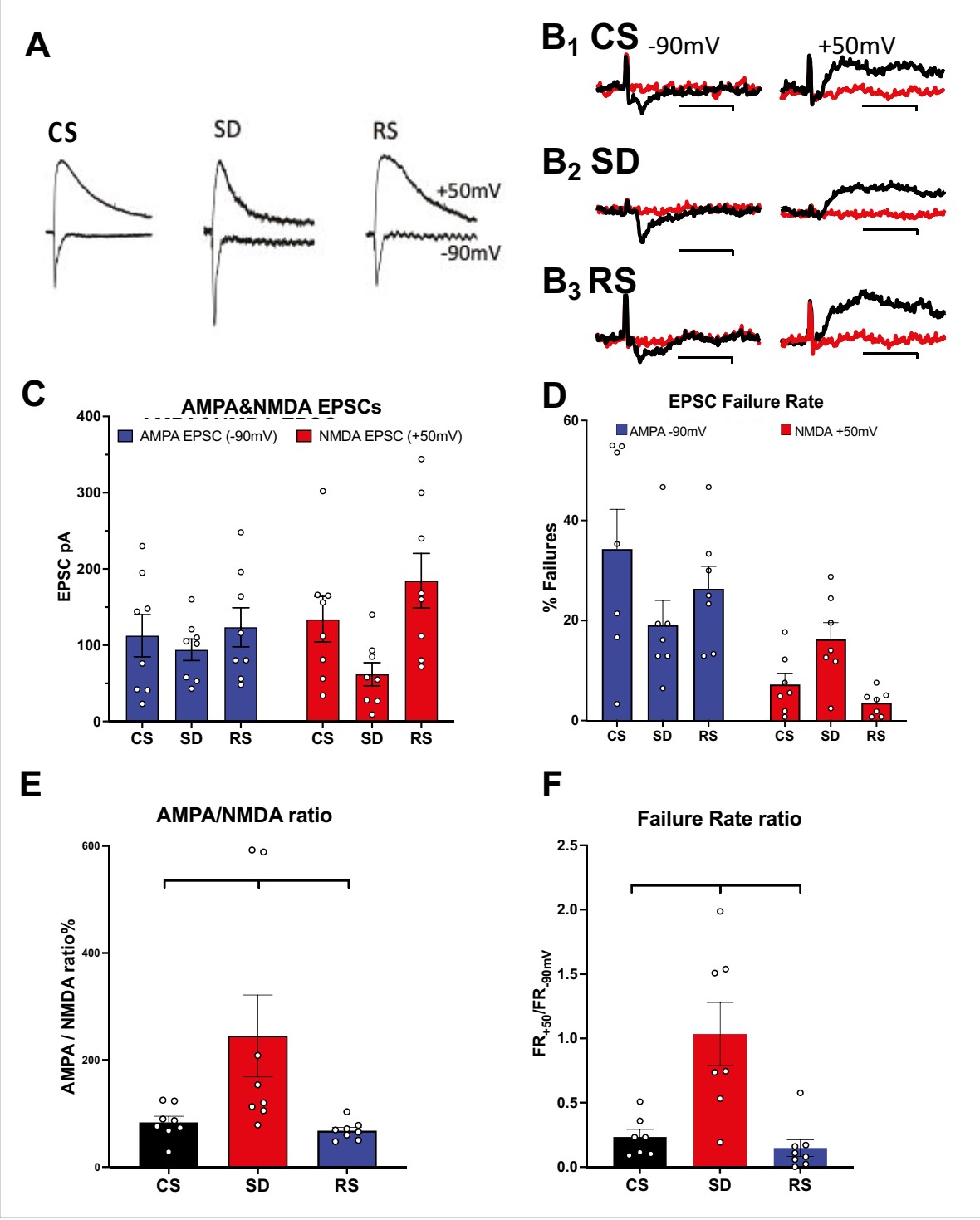

**Figure 1.** Sleep need-dependent responses of AMPA/NMDA ratio and silent synapses in the motor cortex. (**A**) Examples of AMPA currents at –90 mV holding potential and NMDA currents at +50 mV holding potential are shown for: control sleep (CS), 6 hr of sleep deprivation (SD), and 4 hr of sleep deprivation followed by 2 hr of recovery sleep (RS). Traces (100 ms duration) are scaled to the NMDA current for comparison (NMDA current measured @ 40 ms after AMPA peak current). (**B**) Examples of successes (black) and failures (red) at –90 mV (left, AMPA) and +50 mV (right, NMDA) after minimal stimulation of excitatory inputs to motor cortex pyramidal neurons are shown; Top row: CS sleep, middle row: SD and bottom row: RS (Cal. 10pAX20ms). Rate of failures (% of all stimuli delivered) for AMPA EPSCs (blue) and NMDA EPSCs (red) in the three conditions (+/-sem). (**C**, **D**) Average (+/-sem) AMPAR and NMDAR EPSC responses (unmatched) and failure rates, respectively, for each sleep condition. (**E**), (**F**) Matched AMPA/NMDA EPSC response and AMPA/NMDA failure rate ratios, respectively, are shown for the three conditions (N=1 cell/slice, 2–3 slice/animal, 3 animals/condition).

*Figure 1 continued on next page*

eLife Research article

Neuroscience

The online version of this article includes the following source data for figure 1:

**Source data 1.** EPSC subtype amplitudes (Tab 1); statistics for EPSC amplitudes X conditions (Tab2 & 3); failure rate data (Tab4); statistics for failure rate data (Tab5); and statistics ratios of failure rate data (Tab6).

(p<0.916), indicating far fewer numbers of silent synapses (Ss) in this condition. However, both CS and RS restored the number of Ss. $FR_{-90}$ is significantly smaller than $FR_{+50}$ for both CS and RS (p<0.030 and <0.0035, respectively).

The condition-specific ratio of $FR_{+50\,mV}$ to $FR_{-90\,mV}$ (FRR) is a ratio of power functions of the probability of failure for each of the stimulated synapses $(1-p_{release})^{stimulated-synapse}$ and thus, inversely proportional (since $(1-p_{release})<1$) to the number of silent synapses (Ss). A binomial probability distribution model is employed with the assumptions that: (1) each stimulus activates the same set of pre-synaptic terminals for a given neuron; (2) each terminal has a constant $p_{release}$; (3) P(failure) for a given set of stimulus-activated terminals = $(1-p_{release})^{\#activated\ terminals}$. The FRR for $FR_{+50\,mV}/FR_{-90\,mV}$ = $(1-p_{release})^{As+Ss}/(1-p_{release})^{As}$ and $ln(FR_{+50\,mV})/ln(FR_{-90\,mV})$ = (As +Ss)/As = 1 + Ss/As. SD's $FR_{+50\,mV}/FR_{-90\,mV}$=~1, indicating ~no silent synapses. Ss/As for CS = 1.44 and for RS = 1.46, compared to SD = 0.09. As with the AMPA/NMDA RR, we observed a significant difference for FRR between sleep condition cohorts (non-parametric Kruskal-Wallis test, p<0.0006). Significance tests for multiple comparisons between sleep conditions, gave CS-SD, *p*0.02; SD-RS, p<0.0012; and CS- RS, p<0.366 (Two-stage linear step-up procedure of Benjamini, Krieger, and Yekutieli; *Figure 1—source data 1*, tabs 4&6; *Figure 1F*) This suggests there are few if any silent synapses available for conversion by LTP to active synapses following SD and this availability is recovered by sleep.

The sleep-dependent electrophysiological findings raise several questions, including: (1) are other cell types besides the layer 2–3 pyramidal neurons from which we recorded, involved; (2) can changes in the transcriptome account, at least in part, for sleep-related synaptic changes in function, and if so, what are the specific functions of those genes and (3) what is the upstream sleep-dependent control of changes in gene expression?

## Cell type/subtype-specific gene expression in response to sleep loss

To examine the molecular mechanisms responsible for an SD response, we characterized single nuclei transcriptomes of cells from the motor cortex of mice from two cohorts, an ad lib or control sleep (CS) group and an SD group (sleep deprived from ZT = 0–6 hr; see detailed methods).

Transcript libraries from each of the two cohorts (CS, n=4 and SD, n=4), were prepared from single nuclei isolated from mouse frontal cortex. Nuclei from each cohort were clustered and visualized using Uniform Manifold Approximation and Projection (UMAP) and annotated using multi-modal characteristics from the Brain Initiative Cell Census (*BRAIN Initiative Cell Census Network BICCN, 2021*; *Figure 2A and B*). The sleep condition had no significant effect on the expression-based clustering nor on the distribution and numbers of nuclei per cell sub-type across sleep conditions (*Figure 2C*; *Figure 2—source data 1*, tabs 1,2). However, the SD condition was associated with a significantly increased number of transcripts/nucleus across sleep condition, matched for cell subtype (two-way ANOVA, UMI/Nucleus(subType) × Condition(CSvsSD), F (16, 96)=3.004, p=0.0004; Wilcoxon matched-pairs, p≤0.0001; *Figure 2—source data 1*, tab 3; *Figure 2D*). To avoid any technical biases, the CS and SD biological samples were treated in an identical manner, with sacrifice alternating between CS and SD samples, carried out by the same person, brain harvesting and frontal cortex isolation all carried out by Dr. Richa Pandey (blinded as to sample identity), RIN values were indistinguishable and finally, library preparation and sequencing were all completed in a single batch. Thus, the results may be indicative of a generalized increase in transcript number per cell sub-type.

We next determined the differential gene expression (DGE) of transcripts between conditions of CS and SD (see methods; *Figure 3A*). In response to 6 hr of SD from ZT = 0–6, both increased and decreased gene expression was observed in all cell subtypes (*Figure 3—source data 1*). Predictably, there is a trend for increased DEGs in cell types expressing a greater number of genes (*Figure 3A*). The greatest proportion of sleep DEGs is ~69%, expressed by ExIT cell types, while the next largest proportion is ~17% for IN cell types (*Figure 3B*). To determine whether the increased numbers of genes expressed in the ExIT subtypes account for this cell type's high proportion of DEGs, we

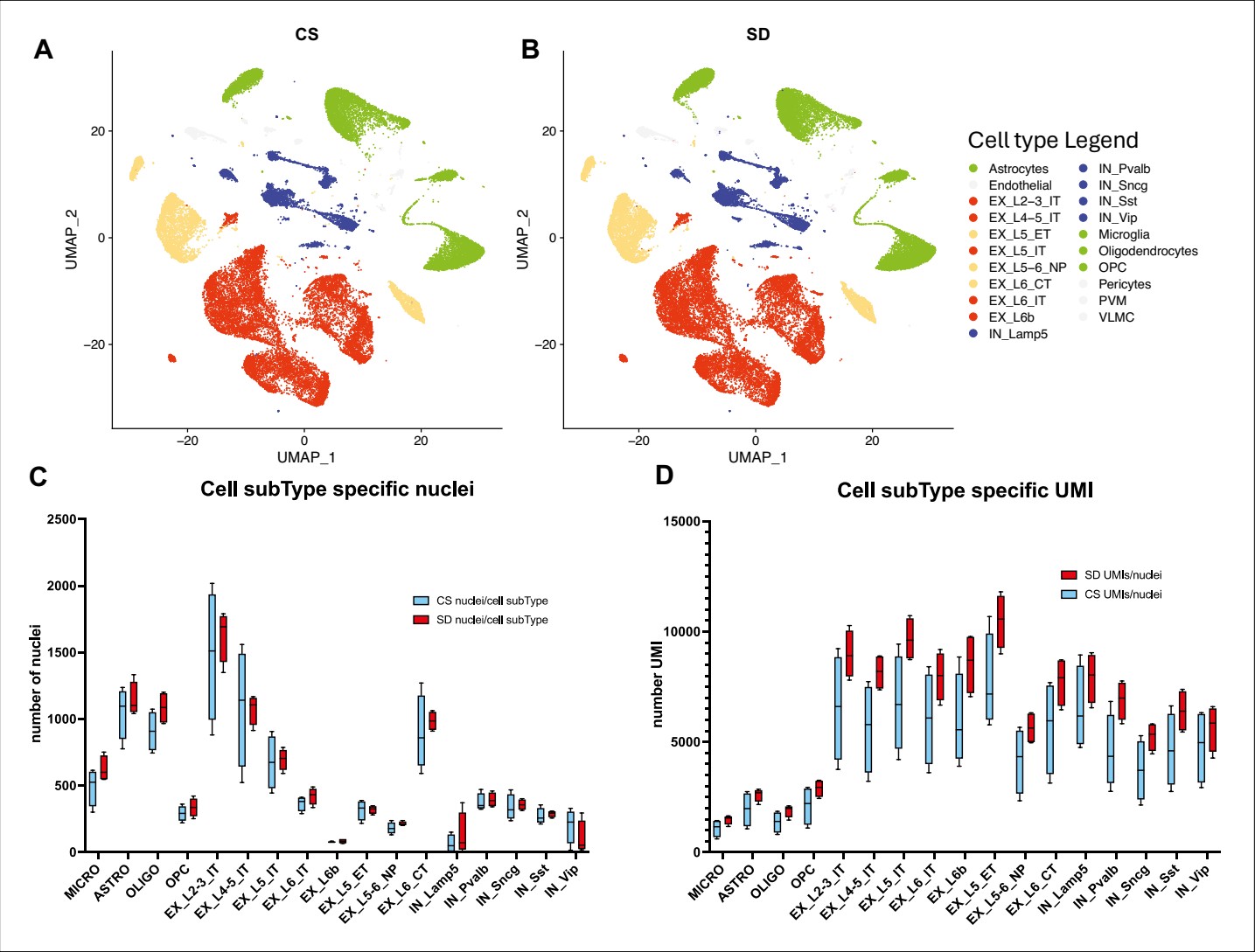

**Figure 2.** snRNAseq data shows cell type and sub-type based on gene expression patterns are unaffected by sleep needs. (**A**) Uniform Manifold Approximation and Projection (UMAP) projection of cell-type gene expression pattern following 6 hr ad lib sleep (control sleep, CS) at ZT = 6 hr. (**B**) As in (**A**) except after 6 hr of sleep deprivation (SD), ZT = 6 hr. (**C**) The distribution of cell numbers across subtypes is unaffected by sleep needs. (**D**) The median number of UMIs/cell is significantly increased by sleep need across all cell subtypes (see *Figure 2—source data 1* and text for statistics).

The online version of this article includes the following source data for figure 2:

**Source data 1.** Table of cell-type specific #'s of cells, genes and UMI's (Tab1); statistics for cell-type specific #'s of nuclei X condition (Tab 2); and statistics for cell-type specific # of UMI's X condition (Tab3).

examined each cell type's probability of DEG expression given the number of genes expressed in the cell type, its Bayesian probability of expression.

The Bayesian probability of sleep loss DEGs, for cell subtype, is more than threefold greater for excitatory intratelencephalically projecting neurons (ExIT subtype, pyramidal cells *BRAIN Initiative Cell Census Network BICCN, 2021*) compared to any other class of cells (*Figure 3C*; *Figure 3—source data 1*, tabs1,2). These observations suggest, with respect to transcriptomic-mediated changes in the frontal cortex, the ExIT class neurons comprise the major target of sleep function and extend earlier observations that frontal glutamatergic neurons are the primary targets of the SD transcriptomic response (*Bjorness et al., 2020*; *Kim et al., 2022*).

We determined the functional relevance of cell-type specific DEGs with respect to both disease risk and biological properties. We found class-selective enrichment of both autism spectrum disorder

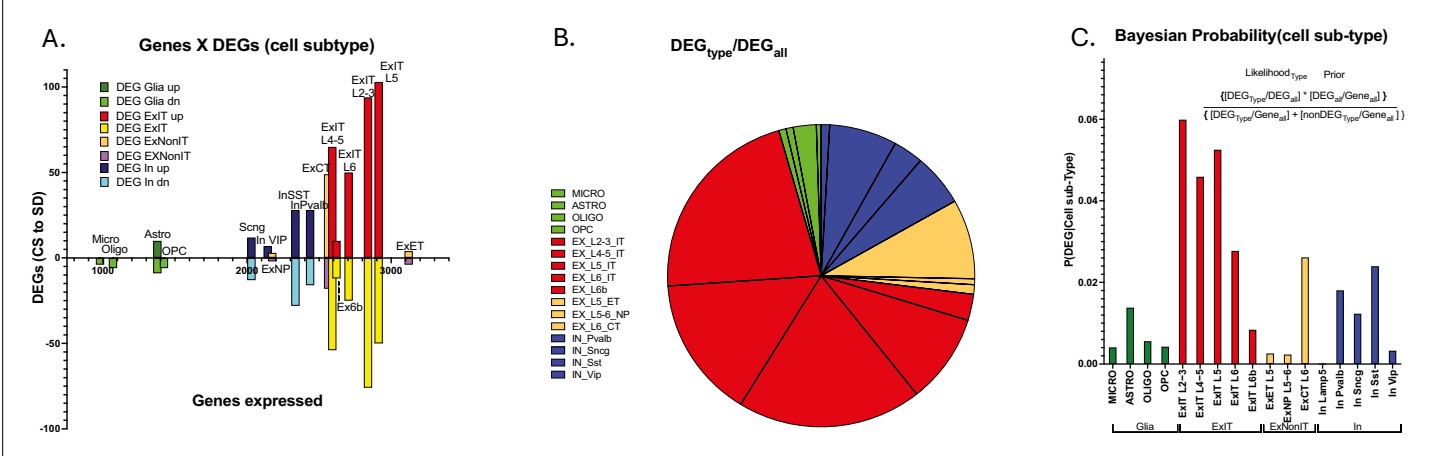

**Figure 3.** Cell type-specific differential gene expression in response to 6 hr of sleep deprivation (SD). (**A**) An XY bar plot of cell type specific differentially expressed genes (DEGs) (both up and down-regulated) organized by cell-type-specific number of expressed genes (X axis) shows the greatest number of DEGs are found in excitatory pyramidal neurons that project within the telencephalon (ExIT). (**B**) A pie chart of the distribution of DEGs amongst different cell types illustrates that the majority of sleep DEGs are expressed by ExIT neurons. (**C**) An analysis of the cell type-specific DEG occurrence shows the greatest probability of significant sleep loss gene response is found in ExIT cells by more than threefold compared to all other cell types.

The online version of this article includes the following source data for figure 3:

**Source data 1.** Table of cell-type and condition specific #'s of genes and DEGs (Tab 1); and cell-type specific distribution of DEGs (Tab 2).

genes (ASD; *Figure 4A*; *Figure 4—source data 1*, tabs 1, 2, 3, 4, 5) and synaptic shaping components (SSC; *Figure 4A*; *Figure 4—source data 1*, tabs 1, 2, 3, 4, 5).

Enrichment of sleep-responsive DEGs by SSC and ASD genes, is strongest for ExIT (L 2–3, 4–5, 5) pyramidal neurons (Chi square with Yates correction p<0.006). In particular, a number of these sleep need ExIT-DEGs encode proteins that can directly affect AMPA/NMDA ratio and silent synapse numbers, including *Dcc* (*Horn et al., 2013*), *Dgkb* (*Kakefuda et al., 2016*), *Gpc6* (*Sato et al., 2016*), *Grin3a* (*Pérez-Otaño et al., 2016*), *Kif17* (*Setou et al., 2000*; *Iwata et al., 2020*), *Kirrl3* (*Martin et al., 2015*), *and Ptprf* (*Sclip and Südhof, 2020*). Some sleep-modulated SSCs are also ASD risk factors in frontal cortical EXIT pyramidal neurons, including *Cdh13, Dcc, Glra2, Gpc6, Grik4, Itpr1, Kirrl3, Nr1d1, Pcdh15,* and *Ush2a*. To a much lesser extent, Layer 6 corticothalamic pyramidal neurons and somatostatin, parvalbumin, and *Sncg* classes of inhibitory neurons (*Scala et al., 2021*) also show this enrichment of their sleep DEGs (*Figure 4A*). These observations are consistent with the ExIT pyramidal cell, glutamatergic, synaptic phenotype as a major functional target for sleep proteins and implicate sleep (dis-)function's role in autism risk.

The transcription factor protein, MEF2c, is necessary for sleep loss gene expression (*Bjorness et al., 2020*). ExIT sleep loss genes are significantly (p<0.0001, Chi$^2$ with Yates correction) enriched with MEF2c target genes (*Harrington et al., 2016*; *Bjorness et al., 2020*; *Figure 4A*.). Class II histone deacetylases 4/5 (referred hereafter as HD4, HD5) are MEF2c binding partners and repressors (*McKinsey et al., 2000*), that can shuttle in and out of the nucleus and are trapped in the cytoplasm when phosphorylated by SIKinases (salt-induced kinases *Kim et al., 2022*; *Zhou et al., 2022*), known to become active in conditions of high sleep need. As the mouse sleep phase progresses HD4/5 is progressively de-phosphorylated in association with decreased sleep need and conversely, as the mouse active phase progresses, HD4/5 is progressively phosphorylated in association with increased sleep need (see Figure 4 in *Zhou et al., 2022*). A virally-mediated expression of a mutant form of *Hdac4*, encoding an HD4 mutant protein that resists phosphorylation and is thus at constitutively high levels in the nucleus (referred to hereafter as HD4cn *Zhou et al., 2022*), is expected to mimic a loss of function of *MEF2c* through constitutive repression of MEF2c. The sleep loss genes observed in frontal cortical ExIT neurons are enriched for both conditional *Mef2c* (*Bjorness et al., 2020*) loss of function DEGs (p<0.0001, Chi$^2$ with Yates correction) and for DEGs in response to expression of HD4cn (*Zhou et al., 2022*) (p<0.0001, Chi$^2$ with Yates correction), as illustrated in a sleep-transcriptome expression model (*Figure 4B*.). Furthermore, these findings are indicative of the role of HD4 as a repressor of MEF2c-facilitated sleep gene expression when sleep need is low and its de-repression of

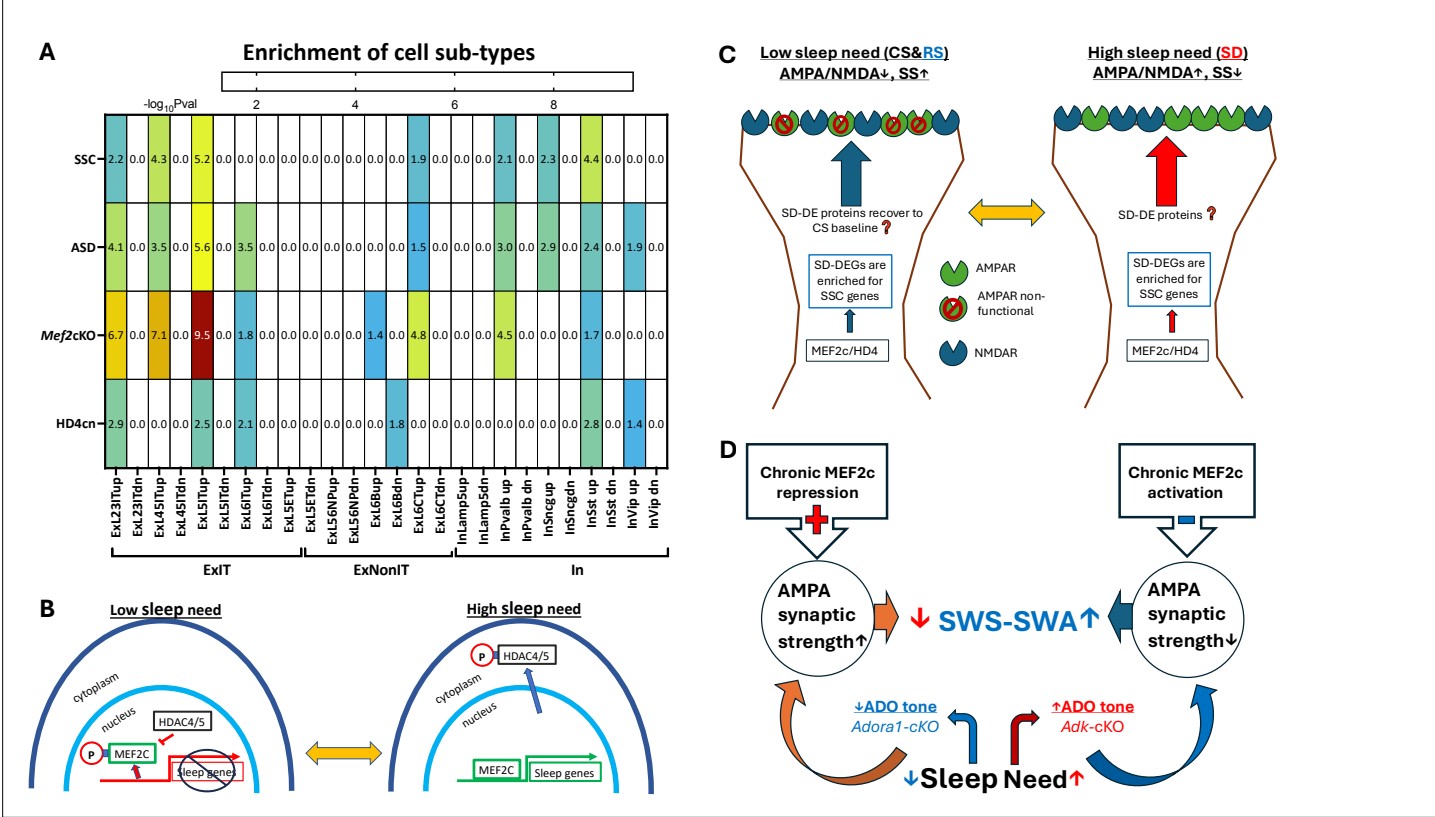

**Figure 4.** Differentially expressed gene (DEG) enrichment of cell types in response to sleep deprivation (SD) by autism risk genes, synaptic shaping component genes, and DEGs from *MEF2c* loss of function or constitutive HD4 repression of MEF2c. (**A**) Heat map for cell type DEG enrichment by autism spectrum disorder (ASD) risk genes, Synaptic Shaping Component genes, *MEF2c*-cKO DEGs, and cnHD4 DEGs. (**B**) Model for the control of sleep DEGs by HD4 repression of MEF2c and by pMEF2c during low sleep need (left). During high sleep need, MEF2c is de-repressed by sequestration of pHD4 to the cytoplasm and dephosphorylation of MEF2c. Both these events facilitate the expression of sleep genes. (**C**) It is proposed that SSC gene expression induced by prolonged waking or SD can, once asleep, decrease the AMPA/NMDA ratio and increase silent synapses (SS) during sleep. This may bias glutamate synapses towards decreased strength but increased potential for long-term plasticity (LTP) in preparation for the active phase, when sleep need is low. Conversely, as the active phase progresses, the AMPA/NMDA ratio increases (as does synaptic strength), and silent synapses are replaced by active synapses in association with increased expression of SSC genes to complete the cycle of glutamate, synapse, and phenotype oscillation. (**D**) An illustration of the slow wave activity during slow wave sleep (SWS-SWA) response to chronic MEF2c repression or activation. Chronic activation of MEF2c facilitated transcription leads to decreased AMPAR-mediated synaptic strength mimicking the effect of increased Ado tone, that will inhibit cortical-thalamic, glutamate synaptic activity to increase SWS-SWA. Chronic repression of MEF2c does the opposite, mimicking loss of Ado A1 receptors (ADORA1) function and tone, to decrease SWS-SWA.

The online version of this article includes the following source data for figure 4:

**Source data 1.** Cell-type specific DEGs from CS to SD, corresponding log2 fold change, and FDR (Tabs 1-4); and curated gene lists and the cell-type specific DEGs (>23% change of expression & FDR < 0.05) that overlapped with one of the curated lists (Tab5).

MEF2c through phosphorylation-induced sequestration into the cytoplasm when sleep need is high (*Zhou et al., 2022*). The resulting high-sleep-need DEGs as noted above, can mediate the observed recovery of glutamate synapse phenotype from a high AMPA/NMDA ratio with sparse silent synapses, to lowered AMPA/NMDA ratio and increased silent synapses (*Figure 4C*).

## Discussion

The major claims of this study are: (1) SD increases the AMPA/NMDA receptor ratio and RS restores it; (2) SD decreases silent synapses compared to CS and RS restores their number after SD; (3) the majority of SD-induced DEGs are found in ExIT cells (glutamate pyramidal neurons projecting within the telencephalon); (4) ExIT SD-induced DEGs are enriched for genes encoding synaptic shaping components and for autism spectrum disorder risk and; (5) these DEGs are also enriched for DEGs

induced by *Mef2c* loss of function restricted to forebrain glutamate neurons (ExIT cells comprise a subset of these) and by over-expression of constitutively nuclear HDAC4 that represses MEF2c transcriptional function. The last claim is consistent with an intracellular signaling model (presented as a hypothesis to be tested, in *Figure 4B*).

Prolonged waking is associated with increased glutamatergic synaptic strength (*Vyazovskiy et al., 2008*; *Liu et al., 2010*; *Bjorness et al., 2020*) and increased number of functional AMPAR synapses (*Bjorness et al., 2020*). Both effects are reversed by recovery sleep (*Bjorness et al., 2020*). The evidence presented here suggests this wake/sleep homeostasis also reflects a waking-induced increase in functional AMPA/NMDA ratio of glutamate synapses of the frontal cortex together with loss of silent glutamatergic synapses. The latter suggests a reduced availability of glutamate synapses for conversion from silent to active, reflecting saturation of LTP. Accordingly, it is reasonable to summarize the implications of sleep-related glutamate synaptic phenotype oscillation as: (1) prolonged waking induces a generalized synaptic strengthening coupled with a negative bias against potentiating plasticity as available slots for AMPARs saturate, and (2) a functional recovery by sleep (*Figure 4C*).

There is evidence that learning of a motor task or experiencing of forced altered motor activity can result in localized increases in NREM (slow wave sleep)-slow wave activity (*Huber et al., 2004*; *Huber et al., 2006*) in the motor cortex. Since SWS-SWA is considered a marker for sleep homeostasis, the altered motor activity-induced increase of SWS-SWA was considered evidence for sleep-related function. Our earlier work has clearly shown that the treadmill method of SD increases frontal cortical SWS-SWA rebound, indicating a sleep-homeostatic process (*Bjorness et al., 2016*; *Bjorness et al., 2020*). Furthermore, we have also shown that this means of experimental SD causes similar glutamate synaptic changes as those observed using other means of SD like gentle handling (*Liu et al., 2010*).

Learning of motor tasks that can be robustly and quantitatively monitored, like bird song in juvenile zebra finches (*Derégnaucourt et al., 2005*; *Kollmorgen et al., 2020*), may reflect the sleep/wake oscillation of bias for LTP and synaptic strengthening. At the start of the active phase, facilitated motor learning improves song performance to a level that appears to saturate by the end of the day. After a night's sleep, performance is degraded but shows enhanced learning, facilitated by decreased AMPA/NMDA ratio and increased availability of synapses that can be readily potentiated. Since there are inevitable, day-to-day, unrelated variances in the environment that should not be learned, mitigation of over-fitting a learned task may require an oscillatory, stairstep-like, learning process. In rats, performance during learning of a novel motor task can display these similar characteristics, especially at the start of learning a novel task when exploration of a multi-dimensional learning space is at a premium (*Kim et al., 2023*), consistent with an underlying sleep/wake oscillation of glutamate synaptic phenotype.

Our unbiased examination of the ontology of ExIT DEGs indicates a response to sleep loss to modify ExIT glutamate synapses involving SSC DEGs. Our electrophysiological observations, now show more particularly, that the gene-expression modifications of ExIT neurons are coupled to SD-responses of, (1) increased AMPA/NMDA ratio; (2) decreased functional silent synapses and; (3) the previously described, increase in synaptic strength and functional number (*Figures 1 and 4*).

The ontology of the ExIT SD-response transcriptome is also notable for enrichment by autism-related risk genes, many of which overlap with our curated category of SSCs. A similar overlap has been noted for social affiliative behaviors, synaptic adhesion molecules, other synaptic shaping molecules with autism/autism spectrum disorder (ASD) risk genes in the mammalian CNS (*Taylor et al., 2020*). The association of sleep disruption with autism is generally appreciated (*Mazurek et al., 2019*) but the focus was on loss of time spent asleep or disrupted sleep, rather than loss of sleep function. The overlapped SSC and ASD DEGs provide a molecular association between sleep's functional role in the motor cortex and ASD risk that captures ASD's association with motor deficits observed in patients (*Chukoskie et al., 2013*) and in mouse autism models (*Cording and Bateup, 2023*).

The molecular mechanisms responsible for sleep/wake DEGs observed in ExIT neurons include two critical transcription factors, MEF2c (*Bjorness et al., 2020*) and HD4/5 (*Kim et al., 2022*; *Zhou et al., 2022*). All neuronal and non-neuronal frontal cortical transcriptomic changes are abolished by loss of function of *Mef2c*. The abolishment of the sleep transcriptome remained even after the restriction of the *Mef2c* knockout to *CamKII*-expressing glutamatergic neurons of the forebrain (*Bjorness et al., 2020*). Importantly, MEF2c was necessary for both the SD-transcriptomic response and for the SD-induced increase of glutamatergic synaptic strength and functional synaptic number as well as for

ensuring sleep-mediated recovery (*Bjorness et al., 2020*). Thus, MEF2c has an essential role in facilitating the expression of sleep genes needed for functional recovery from loss of sleep.

The activity of either of these factors to alter the sleep transcriptome, is controlled by their phosphorylation state. When sleep need is high, MEF2c is de-phosphorylated (*Bjorness et al., 2020*) (from pMEF2c to MEF2c) and Salt Induced Kinases (SIKs) phosphorylate HD4 and HD5 (pHD4/5), sequestering them to the cytoplasm (*Kim et al., 2022*; *Zhou et al., 2022*). Both the phosphorylation of HD4/5 s (*Miska et al., 1999*) and de-phosphorylation of MEF2c (*Zhu and Gulick, 2004*), de-represses MEF2c transcriptional activity. The de-repression of MEF2c and its dephosphorylation are essential for SD-induced differential expression (*Bjorness et al., 2020*). The target genes of MEF2c activation can be inferred from the cortical, differential transcriptome induced by MEF2c loss of function (*Harrington et al., 2016*). These same genes overlap with the differentially expressed genes observed in response to overexpression of constitutively nuclear HD4 (HD4$^{cn}$). This is predictable as HD4$^{cn}$ will constitutively bind and repress MEF2c in the nucleus, thus mimicking a MEF2c loss of function. Furthermore, in association with low sleep need, nuclear, and thus transcriptionally active (as a repressor), HD4, is more abundant in a de-phosphorylated state, relative to transcriptionally inactive, phosphorylated cytoplasmic pHD4 (*Kim et al., 2022*; *Zhou et al., 2022*). NMDAR activation negatively controls HD4 transcriptional activity and nuclear localization (*Sando et al., 2012*). Accordingly, as NMDAR activity accumulates during the active period, an expected accumulation of pHD4 in the cytoplasm occurs to derepress MEF2c activity. Thus, HD4 may act to repress MEF2c transcriptional facilitation of sleep genes, during low sleep need and derepress MEF2c as sleep need builds during the active period. We observed DEGs from both MEF2c loss of function and HD4cn also sleep DEGs consistent with their interaction driven by sleep need (*Figure 4C*).

Both the role of MEF2c to mediate sleep gene expression and nuclear HD4 to repress MEF2c activity, lead to an apparent paradox. When HD4 repression of MEF2c is chronically reduced or lost, sleep need, as indicated by slow wave activity during slow wave sleep (SWS-SWA), is increased (*Kim et al., 2022*). If MEF2c promotes sleep gene expression, then sleep needs should be reduced not increased by its de-repression. Conversely, when constitutively, nuclear HD4$^{cn}$ is overexpressed, to chronically repress MEF2c, sleep need, as indicated by SWS-SWA, is unexpectedly, reduced (*Zhou et al., 2022*). But what is actually being indicated by SWS-SWA?

SWS-SWA has long been employed as a marker for sleep need or intensity, primarily because of its strong correlation with previous time spent awake (*Franken et al., 2001*; *Borbély et al., 2016*). Despite this marker's long history, the mechanisms responsible for its correlation to sleep needs are not well understood.

SWS-SWA correlates with either MEF2c and/or to ADORA1 activation. Either of these activations correlates with a reduction of glutamate synaptic strength (illustrated in *Figure 4D*). Ado activation of Ado A1 receptors (ADORA1) inhibits AMPAR synaptic activity (*Brambilla et al., 2005*; *Greene et al., 2017*). *Adk*, which encodes adenosine kinase in glial cells is the high-affinity metabolizing enzyme of Ado. Glial knockout of *Adk* increases extracellular Ado and SWA in both wake and sleep (*Bjorness et al., 2016*). Rebound SWS-SWA in response to experimental SD increases Ado (*Porkka-Heiskanen et al., 1997*) and the rebound is blocked by a *CamK2a:Cre*-driven conditional knockout of *Adora1* (*Bjorness et al., 2016*). Thus Ado tone, whether increased by prolonged waking activity or by glial knockdown of *Adk*, through its activation of ADORA1, inhibits glutamate synaptic activity, including in arousal centers, to promote SWS-SWA (*Greene et al., 2017*; *Figure 4D*).

Sleep, itself, induces a down-regulation of glutamate synaptic strength, synaptic numbers, and AMPA/NMDA ratio. Chronic facilitation of MEF2c-dependent sleep genes due to de-repression of MEF2c by *Hdac4/5*'s loss of function should induce a chronic down-regulation of glutamatergic synaptic strength and number (*Bjorness et al., 2020*; *Kim et al., 2022*; *Zhou et al., 2022*). The sleep-gene-induced, chronic glutamate synaptic down-regulation can mimic the effect of a tonic increase of extracellular Ado to increase SWA as occurred following glial loss of function of *Adk* (*Bjorness et al., 2016*). Furthermore, synaptic glutamate activity in cholinergic arousal centers (*Rainnie et al., 1994*; *Brambilla et al., 2005*) can be reduced, decreasing cholinergic-mediated thalamo-cortical activation to additionally increase SWS-SWA.

Conversely, repression of MEF2c by de-phosphorylated HD4/5 (or by expression of phospho-dead HD4cn) reduces SWS-SWA (*Zhou et al., 2022*). In this case, glutamate synaptic activity can be chronically increased by the direct effect of loss of sleep function to down-scale glutamate synaptic strength

(*Bjorness et al., 2020*). This can mimic the effect of a loss of *Adora1* function that reduces SWS-SWA (*Bjorness et al., 2009*). It also predicts HD4cn's effect to alter sleep gene expression in response to SD, consistent with the enrichment of SD genes by both HD4cn and *Mef2c*-cKO DEGs, that we observed (*Figure 4A*).

Generally, facilitation or depression of SWA correlates with up or down-scaling effects on cortical glutamate neurotransmission, respectively, even in the absence of a direct effect on sleep need (*Figure 4D*). Many reagents that reduce the excitability of glutamate pyramidal cells by various mechanisms, including anesthetics like isoflurane, barbiturates or benzodiazepines in addition to those activating ADORA1, increase SWA. Those agents that increase excitability of cortical glutamate pyramidal neurons, correlate with decreased SWA. Accordingly, SWA can be a marker for glutamate synaptic strength and number, modulated under normal conditions, by sleep function.

In summary, at the end of a long episode of waking, when sleep needs is high, glutamate synapses of ExIT cells in the frontal cortex are electro-physiologically strengthened, increased in functional number, and show increased AMPA/NMDA receptor ratio, together with decreased availability of silent synapses. The silent synapse conversion by LTP to active synapses is thus limited, creating a bias for increased strength at the expense of potentiating plasticity. In ExIT neuronal nuclei, the transcription factor, MEF2c is de-repressed by de-phosphorylation and sequestration of its co-repressors, the classII HDACs, pHDAC4/5, to the cytoplasm. This facilitates the transcription of sleep genes, including those encoding SSCs controlling glutamatergic synaptic phenotype and ASD risk genes. Recovery sleep recovers the functional phenotype. These observations suggest a daily oscillation from lowered glutamatergic synaptic strength and increased bias for potentiating plasticity at the start of the active phase when sleep need is low, to increased synaptic strength and saturated plasticity at the end of the active phase, when sleep need is high. In conclusion, we have provided electrophysiological evidence for a wake/sleep oscillation of glutamate synapse phenotype that can mediate a glutamatergic strength/meta-plasticity oscillating bias. We also show that this wake/sleep oscillation is likely to occur in glutamate synapses of ExIT pyramidal neurons and be mediated by a select set of synaptic shaping component genes, a significant number of which are also autism risk genes, whose expression is controlled by MEF2c and HD4 transcription factors. Finally, this study implicates a framework within which optimal cortical-dependent motor training can occur in a recursive incremental manner, facilitated by wake/sleep glutamate synapse phenotypical oscillation.

## Materials and methods

### Electrophysiology

All electrophysiological animal experiments were approved by the Animal Experimental Committee at the University of Tsukuba. Animals were housed on a 12:12 hr light/dark cycle (with the light automatically turned on at 9:00) at a stable temperature (24.5 ± 0.5 °C), with free access to food and water. C57/Bl6 male mice aged 8–12 wk were used.

### Slice preparation and recording

Mice were fully anesthetized at ZT 6 with isoflurane, and their brains were promptly removed (<1 min) and placed in ice-cold artificial cerebrospinal fluid (ACSF), composed of 124 mM NaCl, 26 mM NaHCO3, 3 mM KCl, 2 mM CaCl, 1 mM MgSO4, 1.25 mM KH2PO4, 10 mM glucose, 300–310 mOsm, equilibrated with 95% O2 and 5% CO2.

Brains were mounted on the stage of a vibrating microtome (Leica VT1200 S) with cyanoacrylate glue. Coronal sections (300 µm thick) were obtained and allowed to recover in ACSF at room temperature for at least 1 hr. Slices were submerged in a perfusion chamber placed under an upright microscope (BX51WI; Olympus) fitted with a custom LED IR illumination and Nomarski interference contrast. Slices were superfused with ACSF, at a rate of 2 mL/min with Picrotoxin (100 micromolar) added to block GABAA receptor-mediated responses. Neurons in layers 2/3 of the primary motor cortex with a pyramidal cell morphology were patched under visual control. The intracellular solution contained (in mM) 130 Cs gluconate, 1 mM EGTA, 10 mM Hepes, 4 mM MgATP, 0.3 mM NaGTP, and 5 mM NaCl. The pH was adjusted to 7.33 with CsOH and osmolarity to 285–300 mOsm. Borosilicate patch pipettes were pulled to an open-tip resistance of 2–4 MΩ. Signals were amplified and filtered at 10 kHz (Axopatch 700 A; Molecular Devices) and then digitized at 20 kHz using customized routines

in commercial software (IGORPro; WaveMetrics). Series resistance values for the recording pipette ranged between 8 and 15 MOhm and experiments with changes larger than 25% were not used for further analyses.

Evoked synaptic responses were produced through extracellular stimulation (0.1 ms) every 10 s with monopolar glass pipette electrodes filled with ACSF and placed in the vicinity (200–400 micrometers) lateral to the recorded neuron in layer 2/3. For the AMPA/NMDA amplitude ratio experiments stimuli amplitudes were 8.7 µA +/-3.5 µA (SD) and for the failure rate experiments, 1.4 µA +/-0.9 µA (SD).

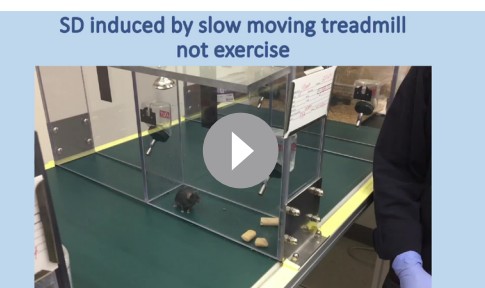

**Video 1.** Video of slow moving treadmill used for sleep deprivation.
https://elifesciences.org/articles/98280/figures#video1

## AMPA/NMDA EPSC analyses

AMPA responses were recorded at a holding potential of –90 mV and mixed AMPA-NMDA responses at a holding potential of +50 mV. The –90 mV holding potential was chosen according to precedent (*Myme et al., 2003*). The hyperpolarized holding potential increases driving force and permits lower stimulus strength for the same response size – reducing the likelihood of polysynaptic responses. Experiments with multiple response peaks at –90 mV were not included in the analysis. The –90 mV holding potential also increases NMDA receptor Mg block - resulting in a minimally contaminated AMPA response.

The AMPA/NMDA ratio was determined according to an established protocol (*Myme et al., 2003*); briefly, at least 10 traces were averaged at –90 mV and +50 mV, the AMPA response was calculated as the peak inward current at –90 mV and the NMDA current was calculated at +50 mV in a 5 ms window 40 ms after the AMPA peak.

Failure rates for AMPA and AMPA +NMDA EPSCs: Minimal stimulation was achieved by lowering the stimulation strength until clear successes and failures could be observed at –90 mV holding potential and stimulation strength was then kept constant for 5–10 min to allow the system to stabilize before failure ratios were determined. Relative numbers of silent and total (silent +active) synapses in an L2-3 pyramidal neuron evoked in a slice by minimal presynaptic stimulation is correlated with failure rates (FR=failures/number of stimuli). $FR - 90 = (1 - p_{release})^{As}$, and $FR + 50 = (1 - p_{release})^{As+Ss}$, where As = number active synapses (at –90 mV) and Ss = number of silent synapses (only active at +50 mV). Since $p_{release}$ for each synapse is the same for –90 mV and +50 mV (i.e. assuming the same probability of failure at both potentials with each stimulus) then:

$$\ln(FR + 50) = \ln(1 - p_{release})^{As+Ss}$$
$$= (As + Ss) * \ln(1 - p_{release})$$
$$\ln(FR - 90) = (1 - p_{release})^{(As)}$$
$$= As^* \ln(1 - p_{release})$$

Thus,

$$\ln(FR + 50)/\ln(FR - 90) = (As + Ss)/As$$
$$= 1 + Ss/As.$$

Since $(FR + 50/FR - 90) = (1 - p_{release})^{As+Ss/(1-p_{release})^{As}}$, as Ss-->0, (FR +50/FR-90)-->1, the maximum ratio possible if prelease is constant. Our uncorrected data show (FR +50/FR-90)=~1.3 in SD, which suggests that FR +50 is overestimated by at least a factor of 1.3 (likely, due to decreased input resistance at Vh = +50 mV). Accordingly, we adjusted FR +50 by this factor = 1.3, for all sleep conditions.

## Sleep deprivation

All mice were accommodated for 1–2 wk in a cage and treadmill, consisting of a bottomless plexiglass enclosure suspended over the treadmill with ad-lib access to food and water. During this time, all

animals were exposed to treadmill movement at ZT = 2 and ZT = 14 hr, for 30 min. For sleep deprivation (SD) the slow-moving treadmill (SMTM; 0.1 KM/h; see *Video 1*) was engaged from ZT 0–6 hr and for recovery sleep (RS) from ZT 0–4. There was no treadmill engagement for RS from ZT 4–6 hr and for control sleep (CS), no treadmill engagement for the entire ZT 0–6 hr period. All animals were sacrificed at ZT = 6 hr and tissue was rapidly prepared for either electrophysiological (exVivo brain slices) or transcriptomic analysis as described below.

## Single nucleus transcriptome analysis

Male 6–10 wk old C57BL/6NCrl (catalog # and substrain) obtained from Charles River Laboratories. All animals for transcriptome analyses were housed within our approved animal facility at the University of Texas Southwestern. Experiments were conducted within approved satellite housing/procedure room within our laboratory space and tissue collection occurred in laboratory spaces approved by the IACUC for this purpose following IACUC-approved procedures: APN 2017–102096, APN 2017–102183.

## snRNA-seq library preparation

Nuclei for snRNA-seq were isolated from male mice, 8–10 wk old. Briefly, the tissue was homogenized using a glass Dounce homogenizer in 2 ml of ice-cold lysis buffer (10 mM Tris-HCl, 10 mM NaCl, 3 mM MgCl2, and 0.1% Nonidet P40 Substitute) and was incubated on ice for 5 min. Nuclei were centrifuged at 500 g for 5 min at 4 °C, washed with 4 ml ice-cold lysis buffer, and incubated on ice for 5 min. Nuclei were centrifuged at 500 g for 5 min at 4 °C. After centrifugation, the nuclei were resuspended in 500 µl of nucleus suspension buffer (NSB) containing 1x PBS, 1% BSA (no. AM2618, Thermo Fisher Scientific) and 0.2 U µl−1 RNAse inhibitor (no. AM2694, Thermo Fisher Scientific). The nucleus suspension was filtered through a 70 µm Flowmi cell strainer (no. H13680-0070, Bel-Art). Debris was removed with density gradient centrifugation using Nuclei PURE 2 M sucrose cushion solution and Nuclei PURE sucrose cushion buffer from the Nuclei PURE prep isolation kit (no. NUC201-1KT, Sigma Aldrich). Nuclei PURE 2 M sucrose cushion solution and Nuclei PURE sucrose cushion buffer were first mixed in a 9:1 ratio. A 500 µl volume of the resulting sucrose solution was added to a 2 ml Eppendorf tube. A 900 µl volume of the sucrose buffer was added to 500 µl of isolated nuclei in NSB. A 1400 µl volume of nucleus suspension was layered on the top of the sucrose buffer. This gradient was centrifuged at 13,000 g for 45 min at 4 °C. The pellet of nuclei was resuspended, washed once in NSB, and filtered through a 70 µm Flowmi cell strainer (no. H13680-0070, Bel-Art). The concentration of nuclei was determined using 0.4% trypan blue (no. 15250061, Thermo Fisher Scientific), and was adjusted to a final concentration of 1000 nuclei per microlitre with NSB.

Droplet-based snRNA-seq libraries were prepared using Chromium Single Cell 3′ v3.1 (1000121, 10 x Genomics) according to the manufacturer's protocol ; *Zheng et al., 2017*. Libraries were sequenced using an Illumina NovaSeq 6000. snRNA-seq preprocessing and annotationRaw sequencing data were obtained as BCL files from the McDermott sequencing core at UT Southwestern. BCL files were demultiplexed using cell ranger mkfastq (10 X Genomics CellRanger suite v3.1.0). Resulting FASTQ files were then accessed for quality using FASTQC (v0.11.5). A reference mouse genome-annotation index was built for mouse genome (GRCm38p6) and Gencode annotation (vM17) using cell ranger mkref (10 X Genomics CellRanger suite v3.1.0). Quality passed FASTQ files were further aligned to reference mouse genome-annotation index and raw count matrices were generated using cellranger count (10 X Genomics CellRanger suite v3.1.0). To remove ambient RNA contamination, we used CellBender's remove-background (https://github.com/broadinstitute/CellBender; *Fleming and Babadi, 2024*) on the raw count matrix per sample. We note that without ambient RNA removal, glial cells were shown to be conspicuously contaminated with neuronal ambient RNAs (*Caglayan et al., 2022*). Also, potential doublets were discarded using the DoubletFinder (https://github.com/chris-mcginnis-ucsf/DoubletFinder; *McGinnis, 2025*) tool. Ambient RNA and doublets cleaned data were used for downstream analysis. For each sample, nuclei with less than 20,000 UMIs and a percentage of reads mapping to mitochondrial genes of less then 0.5 were retained. Individual samples per sleep condition were first integrated using Seurat's (v3) integration approach (IntegrateData) and integrated datasets were further clustered (ScaleData, RunPCA, FindClusters) as described in Seurat's (v3) integration vignette (https://satijalab.org/seurat/archive/v3.0/integration). Clustered data was then visualized (RunUMAP) using Uniform manifold approximation and projection embeddings (UMAP). Cluster-specific gene markers were identified (FindMarkers)

and significant marker genes were enriched using the Fisher exact test for cell-type markers defined in Brain Initiative Cell Census to annotate the cell types. Cell types were also confirmed by expression of canonical marker genes.

Differential gene expression tests were performed using an edgeR-based pseudobulk approach across sleep conditions per cell type. Significant differentially expressed genes (DEGs) were identified using absolute log2 fold change $\geq 0.1375$ (10%) and a false discovery rate (FDR) $\leq 0.05$. Further significant DEGs were enriched across defined gene classes (such as SSC genes, ASD genes, Mef2c-cKO genes and HD4cn genes) using R package SuperExactTest to functionally annotate the genes.

## Statistics

The exact sample size (n) for each experimental group/condition, is provided as a discrete number and unit of measurement. Statistics were taken from distinct samples and not measured repeatedly, with the exception of the matched sampled evoked EPSCs recorded at Vh = −90 mV and at Vh = +50 mV from the same neuron as part of the analyses of AMPAR and NMDAR EPSC amplitudes and failure rates. Statistical details are provided in appropriate tables in tabs labelled 'Stats ....,' in the 'Figure-source data' associated with each figure.

## Additional information

### Competing interests

Genevieve Konopka: Reviewing editor, *eLife*. The other authors declare that no competing interests exist.

### Funding

| Funder | Grant reference number | Author |
| --- | --- | --- |
| NINDS | NS103422 | Robert W Greene |
| NIDOCM | DC014702 | Genevieve Konopka |
| James S. McDonnell Foundation | 10.37717/220020467 | Genevieve Konopka |
| AMEDD | JP21zf0127005 | Kaspar E Vogt |
| International Institute for Integrative Sleep medicine | | Kaspar E Vogt |

The funders had no role in study design, data collection and interpretation, or the decision to submit the work for publication.

### Author contributions

Kaspar E Vogt, Funding acquisition, Investigation, Visualization, Methodology, Writing – review and editing; Ashwinikumar Kulkarni, Formal analysis, Visualization, Methodology; Richa Pandey, Mantre Dehnad, Investigation, Methodology; Genevieve Konopka, Writing – review and editing; Robert W Greene, Conceptualization, Data curation, Formal analysis, Supervision, Investigation, Visualization, Writing - original draft, Writing – review and editing

### Author ORCIDs

Ashwinikumar Kulkarni https://orcid.org/0000-0003-0951-2427
Genevieve Konopka https://orcid.org/0000-0002-3363-7302
Robert W Greene https://orcid.org/0000-0003-1355-9797

### Ethics

This study was performed in strict accordance with the recommendations in the Guide for the Care and Use of Laboratory Animals of the National Institutes of Health. All of the animals were handled according to approved institutional animal care and use committee (IACUC) protocols APN 2017-102096 and APN 2017-102183 @ UTSW medical center. In addition, all electrophysiological animal experiments were approved by the Animal Experimental Committee at the University of Tsukuba, JP.

Reviewer #2 (Public review): https://doi.org/10.7554/eLife.98280.4.sa1
Author response https://doi.org/10.7554/eLife.98280.4.sa2

## Additional files

### Supplementary files
MDAR checklist

### Data availability
Raw and processed data are available at National Center for Biotechnology Information GEO under the accession number GSE256140 at https://www.ncbi.nlm.nih.gov/geo/query/acc.cgi?acc=GSE256140. All analysis scripts are available at https://github.com/konopkalab/sleep_need_seq (copy archived at *Konopka Lab, 2024*).

The following dataset was generated:

| Author(s) | Year | Dataset title | Dataset URL | Database and Identifier |
| --- | --- | --- | --- | --- |
| Vogt K, Kulkarni A, Pandey R, Dehnad M, Konopka G, Greene R | 2025 | Sleep need driven oscillation of glutamate synaptic phenotype | https://www.ncbi.nlm.nih.gov/geo/query/acc.cgi?acc=GSE256140 | NCBI Gene Expression Omnibus, GSE256140 |

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
