## [Editor Report · eLife Assessment]

This **important** study showing that sleep deprivation increases functional synapses while depleting silent synapses supports previous findings that excitatory signaling increases during wakefulness. This manuscript focuses in particular on AMPA/NMDA ratios. An interesting, although speculative, aspect of the manuscript is the inclusion of a model for the accumulation of sleep needs that is based upon the MEF2C transcription factor but also links to the sleep-regulating SIK3-HDAC4/5 pathway. The authors have clarified some questions raised in the previous review, rendering this a **solid** piece of work that poses questions for future studies.

---

## [Referee Report · Reviewer #2 (Public review)]

Summary:

Here Vogt et al., provide new insights into the need for sleep and the molecular and physiological response to sleep loss. The authors expand on their previously published work (Bjorness et al., 2020) and draw from recent advances in the field to propose a neuron-centric molecular model for the accumulation and resolution of sleep need and basis of restorative sleep function. While speculative, the proposed model successfully links important observations in the field and provides a framework to stimulate further research and advances on the molecular basis of sleep function. In my review, I highlight the important advances of this current work, the clear merits of the proposed model, and indicate areas of the model that can serve to stimulate further investigation.

Strengths: Reviewer comment on new data in Vogt et al., 2024

Using classic slice electrophysiology, the authors conclude that wakefulness (sleep deprivation (SD)) drives a potentiation of excitatory glutamate synapses, mediated in large part by "un-silencing" of NMDAR-active synapses to AMPAR-active synapses. Using a modern single nuclear RNAseq approach the authors conclude that SD drives changes in gene expression primarily occurring in glutamatergic neurons. The two experiments combined highlight the accumulation and resolution of sleep need centered on the strength of excitatory synapses onto excitatory neurons. This view is entirely consistent with a large body of extant and emerging literature and provides important direction for future research.

Consistent with prior work, wakefulness/SD drives an LTP-type potentiation of excitatory synaptic strength on principle cortical neurons. It has been proposed that LTP associated with wake, leads to the accumulation of sleep need by increasing neuronal excitability, and by the "saturation" of LTP capacity. This saturation subsequently impairs the capacity for further ongoing learning. This new data provides a satisfying mechanism of this saturation phenomenon by introducing the concept of silent synapses. The new data show that in mice well rested, a substantial number of synapses are "silent", containing an NMDAR component but not AMPARs. Silent synapses provide a type of reservoir for learning in that activity can drive the un-silencing, increasing the number of functional synapses. SD depletes this reservoir of silent synapses to essentially zero, explaining how SD can exhaust learning capacity. Recovery sleep led to restoration of silent synapses, explaining how recovery sleep can renew learning capacity. In their prior work (Bjorness et al., 2020) this group showed that SD drives an increase in mEPSC frequency onto these same cortical neurons, but without a clear change in pre-synaptic release probability, implying a change in the number of functional synapses. This prediction is now born out in this new dataset.

The new snRNAseq dataset indicates the sleep need is primarily seen (at the transcriptional level) in excitatory neurons, consistent with a number of other studies. First, this conclusion is corroborated by two independent, contemporary snRNAseq analysis recently published in iScience 2024 doi: 10.1016/j.isci.2024.110752 and Neuroscience Research 2024 https://doi.org/10.1016/j.neures.2024.03.004. A recently published analysis on the effects of SD in *Drosophila* imaged synapses in every brain region in a cell-type dependent manner (Weiss et al., PNAS 2024), concluding that SD drives brain wide increases in synaptic strength almost exclusively in excitatory neurons. Further, Kim et al., Nature 2022, heavily cited in this work, show that the newly described SIK3-HDAC4/5 pathway promotes sleep depth via excitatory neurons and not inhibitory neurons.

The new experiments provided in Fig1-3 are expertly conducted and presented. This reviewer has no comments of concern regarding the execution and conclusions of these experiments.

Reviewer comment on model in Vogt et al., 2024

To the view of this reviewer the new model proposed by Vogt et al., is an important contribution. The model is not definitively supported by new data, and in this regard should be viewed as a perspective, providing mechanistic links between recent molecular advances, while still leaving areas that need to be addressed in future work. New snRNAseq analysis indicates SD drives expression of synaptic shaping components (SSCs) consistent with the excitatory synapse as a major target for the restorative basis of sleep function. SD induced gene expression is also enriched for autism spectrum disorder (ASD) risk genes. As pointed out by the authors, sleep problems are commonly reported in ASD, but the emphasis has been on sleep amount. This new analysis highlights the need to understand the impact on sleep's functional output (synapses) to fully understand the role of sleep problems in ASD.

Importantly, SD induced gene expression in excitatory neurons overlap with genes regulated by the transcription factor MEF2C and HDAC4/5 (Fig. 4). In their prior work, the authors show loss of MEF2C in excitatory neurons abolished the SD transcriptional response and the functional recovery of synapses from SD by recovery sleep. Recent advances identified HDAC4/5 as major regulators of sleep depth and duration (in excitatory neurons) downstream of the recently identified sleep promoting kinase SIK3. In Zhou et al., and Kim et al., Nature 2022, both groups propose a model whereby "sleep-need" signals from the synapse activate SIK3, which phosphorylates HDAC4/5, driving cytoplasmic targeting, allowing for the de-repression and transcriptional activation of "sleep genes". Prior work shows that HDAC4/5 are repressors of MEF2C. Therefore, the "sleep genes" derepressed by HDAC4/5 may be the same genes activated in response to SD by MEF2C. The new model thereby extends the signaling of sleep need at synapses (through SIK3-HDAC4/5) to the functional output of synaptic recovery by expression of synaptic/sleep genes by MEF2C. The model thereby links aspects of expression of sleep need with the resolution of sleep need by mediating sleep function: synapse renormalization.

Weaknesses:

Areas for further investigation.

In the discussion section Vogt et al., explore the links between excitatory synapse strength, arguably the major target of "sleep function", and NREM slow-wave activity (SWA), the most established marker of sleep need. SIK3-HDAC4/5 have major effects on the "depth" of sleep by regulation NREM-SWA. The effects of MEF2C loss of function on NREM SWA activity are less obvious, but clearly impact the recovery of glutamatergic synapses from SD. The authors point out how adenosine signaling is well established as a mediator of SWA, but the links with adenosine and glutamatergic strength are far from clear. The mechanistic links between SIK3/HDAC4/5, adenosine signaling, and MEF2C, are far from understood. Therefore, the molecular/mechanistic links between a synaptic basis of sleep need and resolution with NREM-SWA activity requires further investigation.

Additional work is also needed to understand the mechanistic links between SIK3-HDAC4/5 signaling and MEF2C activity. The authors point out that constitutively nuclear (cn) HDAC4/5 (acting as a repressor) will mimic MEF2C loss of function. This is reasonable, however, there are notable differences in the reported phenotypes of each. Notably, cnHDAC4/5 suppresses NREM amount and NREM SWA but had no effect on the NREM-SWA increase following SD (Zhou et al., Nature 2022). Loss of MEF2C in CaMKII neurons had no effect on NREM amount and suppressed the increase in NREM-SWA following SD (Bjorness et al., 2020). These instances indicate that cnHDAC4/5 and loss of MEF2C do not exactly match suggesting additional factors are relevant in these phenotypes. Likely HDAC4/5 have functionally important interactions with other transcription factors, and likewise for MEF2C, suggesting areas for future analysis.

One emerging theme may be that the SIK3-HDAC4/5 axis are major regulators of the sleep state, perhaps stabilizing the NREM state once the transition from wakefulness occurs. MEF2C is less involved in regulating sleep per se, and more involved in executing sleep function, by promoting the restorative synaptic modifications to resolve sleep need.

Finally, advances in the roles of the respective SIK3-HDAC4/5 and MEF2C pathways point towards transcription of "sleep genes", as clearly indicated in the model of Fig.4. Clearly more work is needed to understand how the expression of such genes ultimately lead to resolution of sleep need by functional changes at synapses. What are these sleep genes and how do they mechanistically resolve sleep need? Thus, the current work provides a mechanistic framework to stimulate further advances in understanding the molecular basis for sleep need and the restorative basis of sleep function.

Comments on revisions:

No further comments or concerns. I believe that the manuscript has been suitably revised, and the concerns raised by reviewers have been addressed. I am completely satisfied by the revisions and responses provided by the authors.

---

## [Author Response]

The following is the authors’ response to the previous reviews.

**eLife Assessment**
This important study showing that sleep deprivation increases functional synapses while depleting silent synapses supports previous findings that excitatory signaling increases during wakefulness. This manuscript focuses in particular on AMPA/NMDA ratios. An interesting, although speculative, aspect of the manuscript is the inclusion of a model for the accumulation of sleep need that is based upon the MEF2C transcription factor but also links to the sleep-regulating SIK3-HDAC4/5 pathway. The authors have clarified some questions raised in the previous review, but the evidence for major claims was still found to be incomplete, requiring additional experimentation.

The major claims of this study are: (1) SD increases the AMPA/NMDA receptor ratio and RS restores it; (2) SD decreases silent synapses compared to CS and RS restores their number after SD; (3) the majority of SD-induced DEGs are found in ExIT cells (glutamate pyramidal neurons projecting within the telencephalon); (4) ExIT SD-induced DEGs are enriched for genes encoding synaptic shaping components and for autism spectrum disorder risk and; (5) these DEGs are also enriched for DEGs induced by *Mef2c* loss of function restricted to forebrain glutamate neurons (ExIT cells comprise a subset of these) and by over-expression of constitutively nuclear HDAC4 that represses MEF2c transcriptional function. The last claim is consistent with an intracellular signaling model (presented as a hypothesis to be tested, in figure 4B).

[The above is added to the start of the discussion section.]

The specific claims are supported by solid evidence provided in this manuscript. The statistical support is now more clearly presented, with several changes in response to queries by reviewer 1.

The technical issues raised by reviewer 1 do not detract from the claims, thus supported. The rationale for this assessment is expanded below in response to reviewer 1.

Summary:This manuscript by Vogt et al examines how the synaptic composition of AMPA and NMDA receptors changes over sleep and wake states. The authors perform whole-cell patch clamp recordings to quantify changes in silent synapse number across conditions of spontaneous sleep, sleep deprivation, and recovery sleep after deprivation. They also perform single nucleus RNAseq to identify transcriptional changes related to AMPA/NMDA receptor composition following spontaneous sleep and sleep deprivation. The findings of this study are consistent with a decrease in silent synapse number during wakefulness and an increase during sleep. However, these changes cannot be conclusively linked to sleep/wake states. Measurements were performed in motor cortex, and sleep deprivation was achieved by forced locomotion, raising the possibility that recent patterns of neuronal activity, rather than sleep/wake states, are responsible for the observed results.Strengths:This study examines an important question. Glutamatergic synaptic transmission has been a focus of studies in the sleep field, but AMPA receptor function has been the primary target of these studies. Silent synapses, which contain NMDA receptors but lack AMPA receptors, have important functional consequences for the brain. Exploring the role of sleep in regulating silent synapse number is important to understanding the role of sleep in brain function. The electrophysiological approach of measuring the failure rate ratio, supported by AMPA/NMDA ratio measurements, is a rigorous tool to evaluate silent synapse number.The authors also perform snRNAseq to identify genes differentially expressed in the spontaneous sleep and sleep deprivation groups. This analysis reveals an intriguing pattern of upregulated genes controlled by HDAC4 and Mef2c, along with synaptic shaping component genes and genes associated with autism spectrum disorder, across cell types in the sleep deprivation group. This unbiased approach identifies candidate genes for follow-up studies. The finding that ASD-risk genes are differentially expressed during SD also raises the intriguing possibility that normal sleep function is disrupted in ASD.Weaknesses:A major consideration to the interpretation of this study is the use of forced locomotion for sleep deprivation. Measurements are made from motor cortex, and therefore the effects observed could be due to differences in motor activity patterns across groups, rather than lack of sleep per se.

Experimentally induced lack of sleep always involves differences in motor activity. As previously noted in revision 1, motor learning is unlikely to occur in this paradigm and inspection of the video (in supplementary materials) shows no repetitive motor behavioral sequences during the sleep deprivation, nor can this be considered exercise due to the very slow speed of treadmill movement employed. The obvious major difference between groups is a lack of sleep per se. (See below in the “Recommendations for authors”, reviewer 1 for comments on localized wake activity inducing localized sleep-need responses)

Considering that other groups have failed to find a difference in AMPA/NMDA ratio in mice with different spontaneous sleep/wake histories (Bridi et al., Neuron 2020), confirmation of these findings in a different brain region would greatly strengthen the study.

The study of Bridi et al., Neuron 2020, is not comparable to our study for several important reasons. First, their compared groups were from different circadian phases (180 degrees out of phase), whereas in our study, the circadian times for each group were matched (ZT=6hours). Second, experimentally induced sleep loss did not occur whereas it was a focus of our study. Third, spontaneous sleep/wake cannot be accurately matched amongst subjects whereas in our study, sleep loss was matched exactly between groups.

We agree that assessment of AMPA/NMDA ratio and silent synapse number in sleep deprived compared to ad libitum sleep in other areas of the neocortex is of great interest and something we hope to pursue. It would not be surprising to find differences as preliminarily reported by Bahl, et al., Nat Commun. 2024 Jan 26;15(1):779. However, such data would not further strengthen our already well supported evidence for the differences we report in the motor cortex.

The electrophysiological measurements and statistical analyses raise several questions. Input resistance (cutoffs and actual values) are not provided, making it difficult to assess recording quality.

As stated in our first reply, these data were omitted (an admitted oversight on our part) but are now supplied in the methods section as, “Series resistance values for the recording pipette ranged between 8 and 15 MOhm and experiments with changes larger than 25% were not used for further analyses”. We have now also added the Rs/Rm (as a separate column) for each recorded neuron in table 1.

Parametric one-way ANOVAs were used, although the data do not appear to be normally distributed.

We have now removed all the One-way ANOVA tests for clarity (non-parametric tests were previously supplied in addition to the one-way ANOVA tests). Determination of significance with Kruskal-Wallis non-parametric test has not altered statistical support for our conclusions.

Reviewer 1 correctly points out that we had not tested for normality of our distributions- the distributions are likely to be normal but the sample size is too small to confidently make this call for the ratio data which is why we removed the one-way ANOVA’s entirely from table 1.

Two-way ANOVA’s are used to assess AMPA and EPSC amplitudes and failure rates (table 1 tab 2&5) across sleep conditions. As now indicated (table 1, tab 2&5), the distributions of AMPA and NMDA amplitudes and FRs passed the D'Agostino & Pearson test for normality and QQ plots provide illustration supporting this claim.

In addition, for the AMPA/NMDA and FRR measurements (Figures 1E, F), the SD group (rather than the control sleep group) was used as the control group for post-hoc comparisons, but it is unclear why.

The label of “control group” is arbitrary. CS and RS groups are similar (sleep density for RS>CS as expected). Since this appears to be confusing, we now compare all groups to one another in table 1 with the same statistical outcome (additional comparison of CS to RS).

While the data appear in line with the authors' conclusions, the number of mice (3/group) and cells recorded is low, and adding more would better account for inter-animal variability and increase the robustness of the findings.

Of course, the larger the sample, the better the approximation to the population. Our sample sizes yielded significant differences at the usual p<=0.05 threshold with non-parametric testing. A larger sample size could allow for normality testing of the distributions of the data, but fortunately, this was not necessary to support our conclusions.

The snRNAseq data are intriguing. However, several genes relevant to the AMPA/NMDA ratio are mentioned, but the encoded proteins would be expected to have variable effects on AMPA/NMDA receptor trafficking and function, making the model presented in Figure 4C oversimplified. A more thorough discussion of the candidate genes and pathways that are upregulated during sleep deprivation, the spatiotemporal/posttranslational control of protein expression, and their effects on AMPA/NMDA trafficking vs function is warranted.

We have not studied the candidate genes at this point and do not yet understand their potential role(s) in sleep-related AMPA/NMDA functional ratio, only that their expression levels are altered with sleep condition. We agree with the reviewer that the data are intriguing and in need of further investigation. An important first step that can help direct such studies is the identification and preliminary characterization of good candidate genes with respect their cell type specificity, significance and fold change as we have done. Their potential roles likely depend on “the spatiotemporal/posttranslational control” and other factors as reviewer 1 notes.

**Reviewer #2 (Public review):**
Here Vogt et al., provide new insights into the need for sleep and the molecular and physiological response to sleep loss. The authors expand on their previously published work (Bjorness et al., 2020) and draw from recent advances in the field to propose a neuron-centric molecular model for the accumulation and resolution of sleep need and basis of restorative sleep function. While speculative, the proposed model successfully links important observations in the field and provides a framework to stimulate further research and advances on the molecular basis of sleep function. In my review, I highlight the important advances of this current work, the clear merits of the proposed model, and indicate areas of the model that can serve to stimulate further investigation.Strengths:Reviewer comment on new data in Vogt et al., 2024Using classic slice electrophysiology, the authors conclude that wakefulness (sleep deprivation (SD)) drives a potentiation of excitatory glutamate synapses, mediated in large part by "un-silencing" of NMDAR-active synapses to AMPAR-active synapses. Using a modern single nuclear RNAseq approach the authors conclude that SD drives changes in gene expression primarily occurring in glutamatergic neurons. The two experiments combined highlight the accumulation and resolution of sleep need centered on the strength of excitatory synapses onto excitatory neurons. This view is entirely consistent with a large body of extant and emerging literature and provides important direction for future research.Consistent with prior work, wakefulness/SD drives an LTP-type potentiation of excitatory synaptic strength on principle cortical neurons. It has been proposed that LTP associated with wake, leads to the accumulation of sleep need by increasing neuronal excitability, and by the "saturation" of LTP capacity. This saturation subsequently impairs the capacity for further ongoing learning. This new data provides a satisfying mechanism of this saturation phenomenon by introducing the concept of silent synapses. The new data show that in mice well rested, a substantial number of synapses are "silent", containing an NMDAR component but not AMPARs. Silent synapses provide a type of reservoir for learning in that activity can drive the un-silencing, increasing the number of functional synapses. SD depletes this reservoir of silent synapses to essentially zero, explaining how SD can exhaust learning capacity. Recovery sleep led to restoration of silent synapses, explaining how recovery sleep can renew learning capacity. In their prior work (Bjorness et al., 2020) this group showed that SD drives an increase in mEPSC frequency onto these same cortical neurons, but without a clear change in pre-synaptic release probability, implying a change in the number of functional synapses. This prediction is now born out in this new dataset.The new snRNAseq dataset indicates the sleep need is primarily seen (at the transcriptional level) in excitatory neurons, consistent with a number of other studies. First, this conclusion is corroborated by an independent, contemporary snRNAseq analysis recently available as a pre-print (Ford et al., 2023 BioRxiv https://doi.org/10.1101/2023.11.28.569011). A recently published analysis on the effects of SD in *Drosophila* imaged synapses in every brain region in a cell-type dependent manner (Weiss et al., PNAS 2024), concluding that SD drives brain wide increases in synaptic strength almost exclusively in excitatory neurons. Further, Kim et al., Nature 2022, heavily cited in this work, show that the newly described SIK3-HDAC4/5 pathway promotes sleep depth via excitatory neurons and not inhibitory neurons.The new experiments provided in Fig1-3 are expertly conducted and presented. This reviewer has no comments of concern regarding the execution and conclusions of these experiments.Reviewer comment on model in Vogt et al., 2024To the view of this reviewer the new model proposed by Vogt et al., is an important contribution. The model is not definitively supported by new data, and in this regard should be viewed as a perspective, providing mechanistic links between recent molecular advances, while still leaving areas that need to be addressed in future work. New snRNAseq analysis indicates SD drives expression of synaptic shaping components (SSCs) consistent with the excitatory synapse as a major target for the restorative basis of sleep function. SD induced gene expression is also enriched for autism spectrum disorder (ASD) risk genes. As pointed out by the authors, sleep problems are commonly reported in ASD, but the emphasis has been on sleep amount. This new analysis highlights the need to understand the impact on sleep's functional output (synapses) to fully understand the role of sleep problems in ASD.Importantly, SD induced gene expression in excitatory neurons overlap with genes regulated by the transcription factor MEF2C and HDAC4/5 (Fig. 4). In their prior work, the authors show loss of MEF2C in excitatory neurons abolished the SD transcriptional response and the functional recovery of synapses from SD by recovery sleep. Recent advances identified HDAC4/5 as major regulators of sleep depth and duration (in excitatory neurons) downstream of the recently identified sleep promoting kinase SIK3. In Zhou et al., and Kim et al., Nature 2022, both groups propose a model whereby "sleep-need" signals from the synapse activate SIK3, which phosphorylates HDAC4/5, driving cytoplasmic targeting, allowing for the de-repression and transcriptional activation of "sleep genes". Prior work shows that HDAC4/5 are repressors of MEF2C. Therefore, the "sleep genes" derepressed by HDAC4/5 may be the same genes activated in response to SD by MEF2C. The new model thereby extends the signaling of sleep need at synapses (through SIK3-HDAC4/5) to the functional output of synaptic recovery by expression of synaptic/sleep genes by MEF2C. The model thereby links aspects of expression of sleep need with the resolution of sleep need by mediating sleep function: synapse renormalization.Weaknesses:Areas for further investigation.In the discussion section Vogt et al., explore the links between excitatory synapse strength, arguably the major target of "sleep function", and NREM slow-wave activity (SWA), the most established marker of sleep need. SIK3-HDAC4/5 have major effects on the "depth" of sleep by regulating NREM-SWA. The effects of MEF2C loss of function on NREM SWA activity are less obvious, but clearly impact the recovery of glutamatergic synapses from SD. The authors point out how adenosine signaling is well established as a mediator of SWA, but the links with adenosine and glutamatergic strength are far from clear. The mechanistic links between SIK3/HDAC4/5, adenosine signaling, and MEF2C, are far from understood. Therefore, the molecular/mechanistic links between a synaptic basis of sleep need and resolution with NREM-SWA activity require further investigation.Additional work is also needed to understand the mechanistic links between SIK3-HDAC4/5 signaling and MEF2C activity. The authors point out that constitutively nuclear (cn) HDAC4/5 (acting as a repressor) will mimic MEF2C loss of function. This is reasonable, however, there are notable differences in the reported phenotypes of each. Notably, cnHDAC4/5 suppresses NREM amount and NREM SWA but had no effect on the NREM-SWA increase following SD (Zhou et al., Nature 2022).

We speculate that the effect of cnHDAC4/5 to reduce NREM-SWA together with the reduction of NREM amount may be due to a localized increase in neuronal excitability of arousal centers, which would be expected to mask NREM-SWA. Rebound NREM-SWA may reflect the relative rebound increase of NREM-SWA still present under chronic masking conditions (induced by cnHDAC4/5) of increased arousal system excitability. A similar effect to overcome NREM-SWA masking was reported in a Kcna2 KO mouse (a Shaker homologue) by Douglas, et al. (2007, *BMC Biol*).

Loss of MEF2C in CaMKII neurons had no effect on NREM amount and suppressed the increase in NREM-SWA following SD (Bjorness et al., 2020). These instances indicate that cnHDAC4/5 and loss of MEF2C do not exactly match suggesting additional factors are relevant in these phenotypes. Likely HDAC4/5 have functionally important interactions with other transcription factors, and likewise for MEF2C, suggesting areas for future analysis.

This is not a surprising outcome since both MEF2c and HDAC4/5 are transcription factors whose function(s) are determined by multiple other factors a subset of which are relevant to sleep conditions while other determining factors are not necessarily relevant to sleep. These factors can include their phosphorylation state, genomic accessibility, and interaction with other transcription factors. All these other factors are known to be both cell type specific and determined by intracellular conditions, that in turn, are affected by extracellular conditions and ligands. We certainly agree there is much future analysis needed.

One emerging theme may be that the SIK3-HDAC4/5 axis are major regulators of the sleep state, perhaps stabilizing the NREM state once the transition from wakefulness occurs. MEF2C is less involved in regulating sleep per se, and more involved in executing sleep function, by promoting restorative synaptic modifications to resolve sleep need.

A useful way to restate the above might be to distinguish between control of arousal levels determining the behavioral states, wake or sleep (including REM sleep) and control of sleep function. The term, sleep, is typically used to describe the behavioral state of sleep that acts as a permissive gate to sleep function (that resolves sleep need). The sleep state should not be conflated with sleep function. There is abundant evidence that control of arousal can be dissociated from sleep need and sleep function.

Finally, advances in the roles of the respective SIK3-HDAC4/5 and MEF2C pathways point towards transcription of "sleep genes", as clearly indicated in the model of Fig.4. Clearly more work is needed to understand how the expression of such genes ultimately lead to resolution of sleep need by functional changes at synapses.

We are in full agreement. We also note the SIK3-HDAC4/5 pathway may have more than one role, i.e., to affect arousal centers to alter behavioral state and, more generally, to control MEF2c’s transcriptional activity thus controlling sleep-related, glutamate, synaptic phenotype.

What are these sleep genes and how do they mechanistically resolve sleep need? Thus, the current work provides a mechanistic framework to stimulate further advances in understanding the molecular basis for sleep need and the restorative basis of sleep function.
**Recommendations for the authors:**

**Reviewer #1 (Recommendations for the authors):**
Major comments:(1) I appreciate the authors' thoughtful discussion of the use of forced locomotion for their sleep deprivation technique in their response, as well as the additional information that was provided regarding use of the treadmill in the manuscript. However, given that previous studies have failed to find a difference in AMPA/NMDA ratio following spontaneous sleep vs wake, confirmation of the findings in a non-motor brain region with the same SD technique (or confirmation within motor cortex with a different technique, although the authors correctly point out that other techniques also increase locomotor activity) would greatly strengthen the paper.

Addressed above

Notably, differences in motor activity patterns, not necessarily overall amount of locomotion, may induce differential synaptic changes between groups. This point at least warrants acknowledgement and discussion, but this has not been incorporated into the text of the manuscript.

We will incorporate the following into the discussion:

There is evidence that learning of a motor task or experience of forced altered motor activity can result in localized increases in NREM (slow wave sleep)-slow wave activity (Huber R, Ghilardi MF, Massimini M, Tononi G. Local sleep and learning. Nature. 2004;430(6995):78-81); Huber et al., 2006 in the motor cortex. Since SWS-SWA is considered a marker for sleep homeostasis, the altered motor activity induced increase of SWS-SWA was considered evidence for sleep-related function. Our earlier work has clearly shown that the treadmill method of SD increases frontal cortical SWS-SWA rebound, indicating a sleep-homeostatic process (Bjorness et al., 2016; Bjorness et al., 2020). Furthermore, we have also shown that this means of experimental SD causes similar glutamate synaptic changes as those observed using other means of SD like gentle handling (Liu, et al., JoNS 2010).

(2) The number of mice and cells used for electrophysiology in this study remains low; more animals should be included to account for inter-animal variability.

For this study, increasing the number of mice and cells will have p<0.05 chance of altering our conclusions by rejecting the null hypotheses of the electrophysiology findings.

(3) The additional methodological information provided allays some of my concerns regarding the electrophysiological data. However, information about the input resistance (cutoffs used and/or actual values) is still not provided, which is important for assessing recording quality.

We have now supplied the experimentally determined input resistance for each neuron used in this study (a separate column in table 1, tabs marked, “data”).

(4) It is not meaningful to compare raw AMPA or NMDA responses because stimulus electrode placement will differ between cells, potentially activating different numbers of afferents. Presenting these comparisons (Figure 1C) has the potential to mislead the reader.

This is not misleading (it didn’t mislead reviewer 1) as we described the conditions. As expected by reviewer 1, the variability using “raw AMPA or NMDA responses…” was too great, but did indicate an interaction between receptor responses and sleep condition. This provided (as stated in the results section) rationale to examine, and to only draw conclusions from the AMPA/NMDA amplitude and FR ratios.

(5) I appreciate clarification on the statistics and the authors' response has answered some of my questions. However, this also raises additional questions. What test was used to determine normality (and therefore whether to perform a parametric vs nonparametrictest)?

Described above.

Why was the FRR data analysis changed to a parametric test, when it does not appear that the data are normally distributed?

Showing the parametric test was a mistake on our part- there are not enough samples to conclusively conclude the distributions are normal as reviewer 1 correctly suspects. However, the non-parametric Kruskal-Wallis tests that we also show in table 1 indicate significant differences between conditions and the non-parametric, two-stage linear step-up procedure of Benjamini, Krieger and Yekutieli, indicates significant differences between CS-SD and RS-SD but not for CS-RS, supporting our conclusions. The (unsupported) parametric tests are now removed in Table 1 leaving behind the non-parametric test.

Why were post-hoc tests chosen to compare to a control group rather than all pairwise comparisons,

We now provide post-hoc all-pairwise comparisons to give the same results using the BKY analysis.

and why was the SD rather than CS group used as the control in Figures 1E and F?Why were different post-hoc tests chosen for the data in Figures 1E, F?

There was no need for this and we now, only show statistics that are used to draw our conclusions for the AMPA/NMDA EPSC ratios data shown in Figure 1E and Failure Rate Ratios data shown in Figure 1F (the conclusions are supported by the non-parametric post-hoc test and remain unchanged).

(6) Genes in the SSC, ASD, Mef2cKO, and HD4cn categories are almost exclusively upregulated in the SD group compared to the CS group (Figure 4A). As the authors point out in their response, "No claim of mechanism linking the changed expression to altered AMPAR or NMDAR activity can be made at this point," largely due to the fact that we do not know the spatiotemporal or posttranslational modification patterns of the translated proteins, and how they affect receptor trafficking vs function. This is in agreement with my original point: as written (and as illustrated in Figure 4C), the manuscript implies that upregulation during SD increases the AMPA/NMDA ratio via receptor trafficking,

The model indicates a likely (but not necessarily exclusive) role for AMPA/NMDA trafficking to explain the functional electrophysiological data that we do report and which is not in dispute. The SSC-DEGs in ExIT cells are consistent with sleep-altered AMPA/NMDA trafficking but remain only a correlation. However, the point is taken and Figure 4c has been revised to only reflect what we have observed electrophysiologically and the speculated mechanism(s) mediated by observed SSC-DEGs are illustrated with “?’s”.

while in reality the picture is likely much more complicated, and therefore a more thorough discussion is warranted. Some discussion was provided in the authors' response but does not appear to have been incorporated into the text or Figure 4C.

As indicated above the proposed model is changed in Figure 4c to more explicitly indicate which aspects reflect our electrophysiological data and which aspects reflect only an association of observations.

Minor comments:(1) Please justify only using male mice

We had to start somewhere with our limited resources. Our intentions are to follow up with similar experiments using female mice, should funding be realized.

(2) The model in Figure 4C is oversimplified and remains problematic, for the reasons stated in comment #6, above.

See responses above.

(3) Figure 4D remains confusing

We agree. The unnecessary addition of adenosine effects on cholinergic arousal centers (experimentally well supported), have been removed from the figure to provide a more focused indication of how SWS-SWA can be related to either MEF2c and/or to ADORA1 activation through reduction of glutamate synaptic strength. ADORA1 activation elicits reduced glutamate synaptic activity through pre- and postsynaptic inhibition whereas MEF2c activation is essential to reduce sleep elicited, glutamate EPSC reduction. Reduced glutamate synaptic strength, whatever the cause, is associated with increased SWS-SWA.